**DOI: 10.1038/ncomms14119**　　**OPEN**

# Correlation-driven transport asymmetries through coupled spins in a tunnel junction

Matthias Muenks[1], Peter Jacobson[1], Markus Ternes[1] & Klaus Kern[1,2]

Spin–spin correlations can be the driving force that favours certain ground states and are key in numerous models that describe the behaviour of strongly correlated materials. While the sum of collective correlations usually lead to a macroscopically measurable change in properties, a direct quantification of correlations in atomic scale systems is difficult. Here we determine the correlations between a strongly hybridized spin impurity on the tip of a scanning tunnelling microscope and its electron bath by varying the coupling to a second spin impurity weakly hybridized to the sample surface. Electronic transport through these coupled spins reveals an asymmetry in the differential conductance reminiscent of spin-polarized transport in a magnetic field. We show that at zero field, this asymmetry can be controlled by the coupling strength and is related to either ferromagnetic or antiferromagnetic spin–spin correlations in the tip.

[1] Max Planck Institute for Solid State Research, Heisenbergstrasse 1, 70569 Stuttgart, Germany. [2] Institut de Physique, École Polytechnique Fédérale de Lausanne, 1015 Lausanne, Switzerland. Correspondence and requests for materials should be addressed to M.T. (email: m.ternes@fkf.mpg.de).

Correlation is a fundamental statistical measure of order in interacting quantum systems. In solids, electron correlations govern a diverse array of material classes and phenomena such as heavy fermion compounds, Hunds metals, high-$T_c$ superconductors, and the Kondo effect[1–5]. Spin–spin correlations, notably investigated by Kaufman and Onsager in the 1940s (ref. 6), are at the foundation of numerous theoretical models but are challenging to measure experimentally. Reciprocal space methods can map correlations[7], but at the single atom limit new experimental probes are needed. Using the scanning tunnelling microscope (STM) as a manipulation tool, it is possible to construct atomically precise magnetic nanostructures and explore the exchange interaction between neighbouring spins on surfaces[8–10]. For example, the Ruderman–Kittel–Kasuya–Yosida interaction, an oscillatory exchange mechanism, has been observed for pairs of spins on magnetically susceptible platinum surfaces and Neel states have been engineered in antiferromagnetically coupled arrays[11,12]. Similarly, the global consequences of correlation, such as the superconducting gap or zero bias anomalies due to the Kondo effect, have been found and explored in STM experiments[13,14]. Competing energy scales, a telltale sign of strongly correlated systems, have recently come under investigation in the two-impurity Kondo problem and the coupling of magnetic molecules to superconducting hosts[15–17]. Even with these successes, direct measurements of correlation in nanomagnetic systems have proven elusive[18]. To directly determine spin–spin correlations, transport experiments through coupled spins, much in the same manner as coupled mesoscopic quantum dots[19–22], can be performed with the STM.

Here, we use local spectroscopy to study electronic transport through such a coupled spin system. Each metallic lead, tip and sample, harbours an atomic spin system enabling the coupling between the two spins to be smoothly controlled by varying the tip-sample separation. Our coupled spin system is intrinsically asymmetric; the spin bound to the tip is strongly hybridized with the bulk Pt metal and spectroscopically unremarkable, while the spin at the sample surface is decoupled from the underlying Rh metal by an insulating $h$-BN monolayer leading to strong spectroscopic signatures. The transport characteristics through this junction show distinctive asymmetries in the differential conductance ($dI/dV$), that are a direct result of spin–spin correlations between the strongly hybridized atomic spin on the tip apex and its surrounding electron bath in the tip metal electrode. By taking these correlations into account, we can describe and model the observed asymmetries within an electronic transport model. We find correlations up to 60% between the state of the spin system on the tip and the itinerant bath electrons of the tip.

## Results

**Experimental outline**. Figure 1a sketches our experiment, in which we probe a CoH complex on the $h$-BN/Rh(111) sample surface[23]. Using vertical atom manipulation[24], we functionalize our initially bare tip apex with a Co atom (Fig. 1b, see methods) and subsequently probe a second CoH complex (Fig. 1c,d). For the Co-functionalized tip apex, we observe significant changes in the $dI/dV$ spectra when we vary the conductance setpoint, $G_s$. Note that magnetic adatoms on Pt surfaces are subject to strong hybridization with the substrate, making it difficult to determine the spin state using local spectroscopy[25–28]. In our energy range of interest, bare Pt as well as Co-functionalized tips are spectroscopically nondescriptive[29] (see Supplementary Fig. 1). We describe the Co-functionalized tips as a half-integer spin system that is strongly interacting

with the electrons of the Pt tip while remaining spectroscopically dark.

**Spectroscopic measurements**. For a detailed look at the change of the $dI/dV$ spectra, we incrementally increase $G_s$. Figure 1e shows the result for the non-functionalized, that is, a bare Pt tip. The spectra are characteristic for a $S = 1$ spin system with magnetic anisotropy and no level degeneracy, as shown in our earlier work[23]. We observe step-like increases in the $dI/dV$ signal due to current-induced transitions from the ground state to the two excited states (Fig. 1f). The energetic position of these transitions does not change when $G_s$ is increased by more than an order of magnitude. However, by employing Co-functionalized tips and increasing $G_s$ over a similar range to the bare tip, the step positions shift to higher energies and a conductance asymmetry appears at the energetically higher (outer) step. Two prototypical sets of spectra measured on two similar CoH complexes but with two different Co-functionalized tips are shown in Fig. 1g,h. Apart from slightly different excitation energies due to the $h$-BN corrugation that influences the magnetic anisotropy of the CoH complexes on the sample[23,30], these two sets vary in their $dI/dV$ asymmetry at high $G_s$. The data in Fig. 1g show higher $dI/dV$ at negative bias, while the spectra in Fig. 1h show the opposite trend with an enhanced $dI/dV$ at positive bias.

To quantify these changes, we determine the step energies and the $dI/dV$ asymmetries, $\eta_i$ and $\eta_o$, of the inner and outer steps, respectively, for different $G_s$ (Fig. 2). The asymmetry, $\eta = (h_n - h_p)/(h_n + h_p)$, is defined by the intensity of the steps at negative, $h_n$, and positive voltages, $h_p$ (refs 31,32). Spectra obtained with Co-functionalized tips at high $G_s$ show an evolution of the step energies reminiscent of those produced by Zeeman splitting in an external magnetic field oriented along the surface normal[23]. Likewise, the asymmetries resemble spectra obtained with a spin-polarized tip in an external magnetic field[31,32]. However, the changes observed here occur in the absence of an external magnetic field and only as $G_s$ is increased.

**Model Hamiltonian**. To model these results, we employ a spin Hamiltonian that includes axial, $D$, and transverse magnetic anisotropy, $E$, for the $S_1 = 1$ CoH spin adsorbed on the $h$-BN/Rh(111) sample substrate. Similar to earlier experiments[23], we find easy-axis anisotropy, $D < 0$, which favours states with high magnetic moments, $m_z = |\pm 1\rangle$. The non-negligible $E$ term leads to non-magnetic superpositions[33]: an antisymmetric ground state, $\frac{1}{\sqrt{2}}(|-1\rangle - |+1\rangle)$, and a symmetric first excited state, $\frac{1}{\sqrt{2}}(|-1\rangle + |+1\rangle)$ (Fig. 1e). To account for the functionalized tip, we add a term that explicitly describes the direct exchange coupling between the spin on the sample, $S_1$, and the tip, $S_2$:

$$H_0 = D\hat{S}_{1,z}^2 + E(\hat{S}_{1,x}^2 - \hat{S}_{1,y}^2) + J_{12}\hat{S}_1 \cdot \hat{S}_2 + \sum_{i=1}^{2} g_i \mu_B B_z \cdot \hat{S}_{i,z}, \quad (1)$$

where $\hat{S}_i = (\hat{S}_{i,x}, \hat{S}_{i,y}, \hat{S}_{i,z})$ are the corresponding spin operators of the $i$-th spin and $J_{12}$ is the coupling between the two spin systems. The effect of an external magnetic field, $B_z$, is accounted for by Zeeman terms that include the gyromagnetic factor for each spin, $g_i$, and the Bohr magnetron, $\mu_B$.

We approximate the Co-functionalized tips as $S_2 = 1/2$ and diagonalize $H_0$ yielding six eigenstates, $|\psi_k\rangle$, which are twofold degenerate at $B_z = 0$ T. Surprisingly, this simple model enables us to fit the evolution of the step energies when we assume that the coupling between the two spins on tip and sample is either Heisenberg-like, $J_{12} = (J_{12}, J_{12}, J_{12})$, (Fig. 2c) or Ising-like, $J_{12} = (0, 0, J_{12})$, (Fig. 2e). We find that the direct exchange

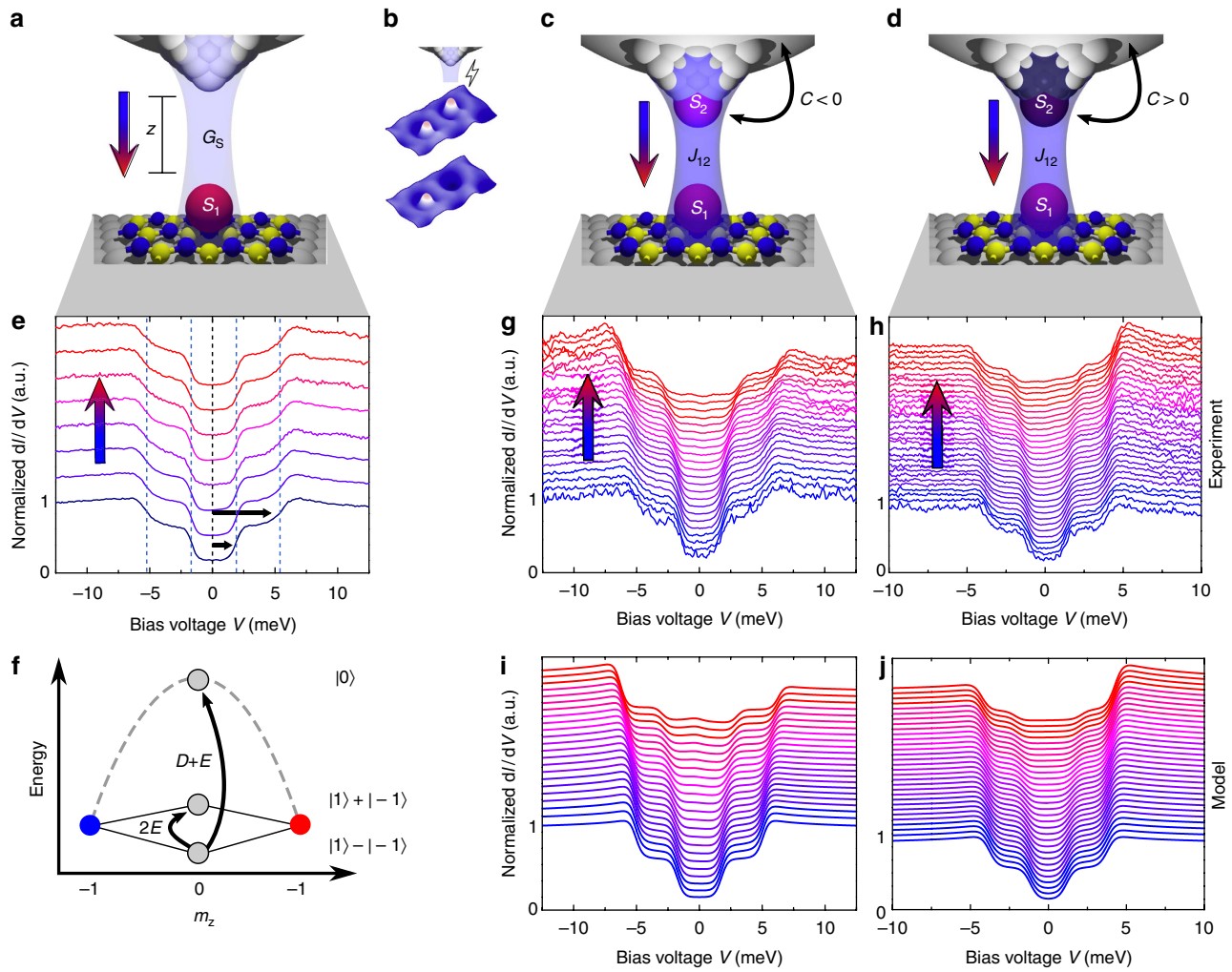

**Figure 1 | Schematics of the experiment and differential conductance.** (**a**) Schematic of the tunnel junction with a Pt tip (top) and a Rh sample (bottom), harbouring a CoH complex (red sphere) with spin $S_1 = 1$ that is decoupled by a monolayer of $h$-BN (B yellow, N blue spheres). The width, $z$, can be smoothly tuned by the conductance setpoint $G_s = I_s/V_s$ (blue-red arrow). (**b**) 3D representations of two successively recorded constant current topographies (size $3 \times 7$ nm$^2$, $V_s = -200$ mV, $I_s = 20$ pA, $T = 1.4$ K) illustrating the transfer of a Co atom to the tip by vertical atom manipulation. (**c**) Schematics of a Co atom on the tip apex that we approximate with spin $S_2 = 1/2$. Hybridization with the Pt tip leads to an antiferromagnetic correlation, $C < 0$, between the Co spin and the spin of the tip electrons. The spins $S_1$ and $S_2$ are exchange coupled by $J_{12}$ that is proportional to $G_s$. (**d**) Schematic of a different tip that shows ferromagnetic correlation, $C > 0$. (**e**) Evolution of the differential conductance (d$I$/d$V$) spectra obtained with a bare Pt tip as illustrated in **a** when increasing $G_s$ from 33 nS (blue curve) to 466 nS (red curve). Black vertical arrows indicate the two excitation steps at around $\pm 2$ and $\pm 5$ meV. (**f**) State diagram of the CoH $S = 1$ in which axial ($D$) and transverse anisotropy ($E$) lift the degeneracy. The magnetic ground and first excited state are non-magnetic superpositions of the $m_z = +1$ and $-1$ states. The arrows depict the two transitions leading to the excitation steps in **e**. (**g,h**) Evolution of d$I$/d$V$ spectra recorded with the two different Co-functionalized tips as illustrated in **c** and **d**. For low $G_s$ (blue), both sets are similar to the spectra in **a**. As $G_s$ increases (red), a change of the excitation energy occurs and the step at higher energy becomes asymmetric. (**i,j**) Simulations reproducing the data in **g** and **h**. Spectra are vertically offset for better visualization.

coupling, $J_{12}$, is proportional to the conductance, $G_s$, and both are an exponential function of the distance $z$ between the two spins[34], allowing us to exclude the magnetic dipolar interaction. Note that we cannot determine the absolute value of the $S_2$ spin. Assuming $S_2 = 3/2$ leads to similar results when changing the proportionality between $J_{12}$ and $G_s$ accordingly. We describe this orbital overlap between the tip and sample spin with an anti-ferromagnetic (AFM) coupling, $J_{12} > 0$, which we will justify further below. The principle evolution of these six eigenstates with $J_{12}$ is shown in Fig. 3. For $J_{12} \to 0$ (Fig. 3b), the combined spin system can be described as a set of doublets, which are only an expansion of the single CoH spin shown in Fig. 1f. Increasing $J_{12}$ not only leads to higher excited state energies of the excited states, but also to a clear separation in states with different total

magnetic moment in $z$-direction, $m_z^t = \langle \hat{S}_{1,z} \rangle + \langle \hat{S}_{2,z} \rangle$, similar to spintronic magnetic anisotropy[35]. In addition, the coupling results in a concomitant polarization, $\langle \hat{S}_{1,z} \rangle$, of the $S = 1$ subsystem, counteracting the $m_{1,z} = |-1\rangle$ and $|+1\rangle$ superposition of the four energetically lowest states. Here an exchange coupling of $J_{12} = 2$ meV is sufficient to polarize the ground and first excited states in the doublets with weights greater than 0.85 (Fig. 3c). However, with no external magnetic field to break the degeneracy of the doublets, the time-averaged magnetization of the spin system remains zero.

**Electrical transport**. We now continue to describe the electrical transport through the junction by employing an interaction

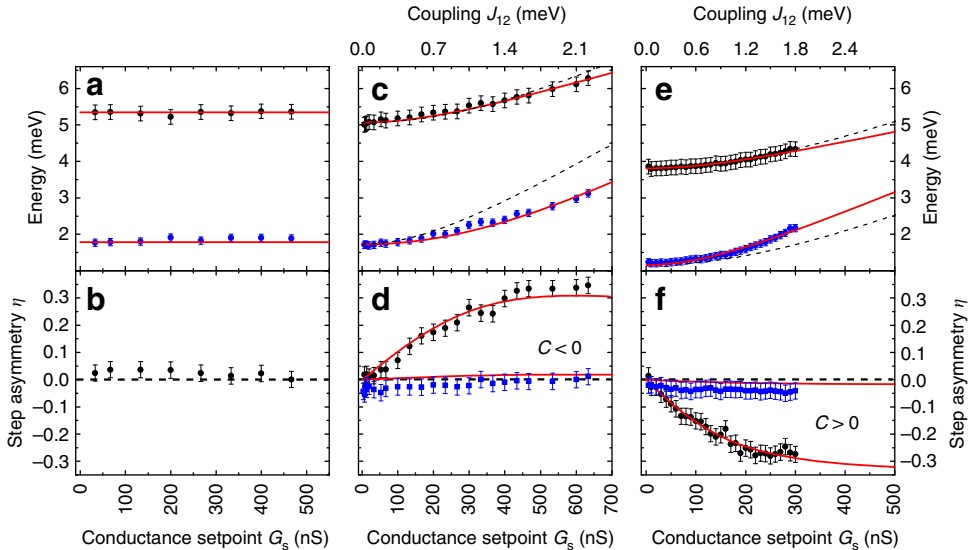

**Figure 2 | Fits to the transport model.** (**a–f**) Evolution of the experimentally obtained step energies and asymmetries $\eta$ (black dots outer steps, blue dots inner steps) together with least-square fits to our model (red lines). (**a,b**) For the bare Pt tip, the excitation energies remain constant and $\eta \approx 0$. (**c**) For the antiferromagnetic Co-functionalized tip, we find $D = -4.2$ meV, $E = 0.87$ meV and a Heisenberg-like coupling between the two spins $S_1$ and $S_2$ with a strength of $J_{12} = 3.6$ µeV/nS × $G_s$ describing the evolution of the excitation energies best. (**d**) An additional correlation of $C = -0.5 \pm 0.05$ between $S_2$ and the tip electrons fits the $\eta$. (**e**) Ferromagnetic Co-functionalized tip fits with the parameters $D = -3.23$ meV, $E = 0.58$ meV and an Ising-like coupling $J_{12} = 5.9$ µeV/nS × $G_s$. (**f**) For this tip, a correlation of $C = 0.35 \pm 0.04$ fits $\eta$. The error bars include the statistical and systematic error of the fit to the spectroscopic data. In particular, for the energy positions, the error is one standard deviation of the Gaussian fit to the first derivative of the spectroscopic data and for the asymmetry proportional to the areas under the Gaussian fits. For comparison Ising (**c**) and Heisenberg fits (**e**) are shown (dashed line), not reproducing the experimental obtained step energies.

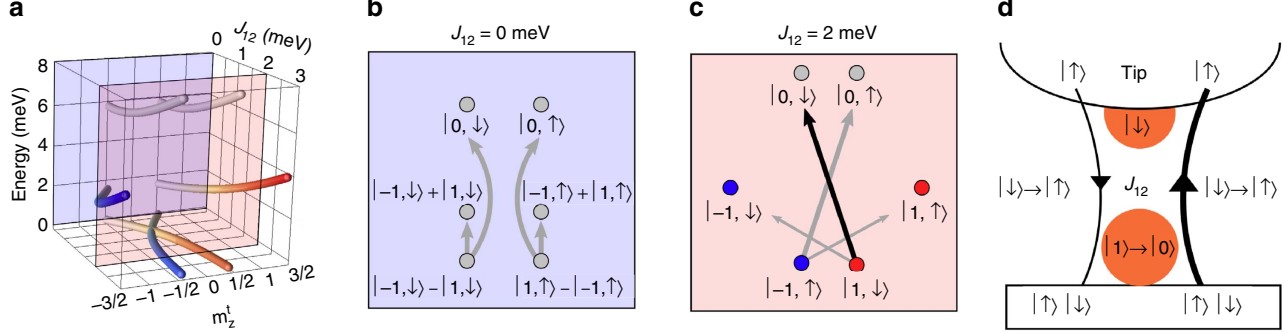

**Figure 3 | State evolution of the combined spin system.** (**a**) Simulated evolution of the state energies and the total magnetic moments, $m_z^t$, of the combined spin system, $S_1 = 1$ and $S_2 = 1/2$, with Heisenberg coupling $J_{12}$. The colour code shows the projected magnetic moment $\langle S_{1,z} \rangle$ of the $S_1$ subsystem (blue: $-1$, red: $+1$). (**b**) 2D cut at $J_{12} = 0$ meV. Grey arrows show the main excitation channels from the two degenerate ground states. This scheme is a doublet of the one shown in Fig. 1f due to the additional spin degree of $S_2$. (**c**) 2D cut at $J_{12} = 2$ meV. The coupling introduces a strong polarization of the states. Angular momentum conservation only allows $\Delta m_z = 0, \pm 1$ transitions, suppressing the excitation to the energetically lower states (thin grey arrows). The transitions to the energetically higher states strongly depends on the spin distributions in tip and sample as well as on the tunnelling direction. (**d**) Illustration of the origin of the bias asymmetry for the transition depicted as black arrow in **c**.

Hamiltonian, $\frac{1}{2}\hat{\boldsymbol{\sigma}} \cdot \hat{\boldsymbol{S}} + U$, between the tunnelling electrons and the coupled spin system, with $\hat{\boldsymbol{\sigma}} = (\hat{\sigma}_x, \hat{\sigma}_y, \hat{\sigma}_z)$ as the standard Pauli matrices and $\hat{\boldsymbol{S}} = \hat{\boldsymbol{S}}_1 \otimes \hat{\boldsymbol{S}}_2$ as the combined spin operator of the two spins. The first term of this interaction describes Kondo-like spin-flip scattering processes, while the second term describes spin-conserving potential scattering processes[24,36]. The potential scattering processes have only marginal influence on the spectrum and, in particular, on the asymmetry $\eta_o$ (see Supplementary Fig. 2). Therefore, we will neglect it in the following by setting $U = 0$. The systematic offset between observed and calculated $\eta_i$ could be due to a non-zero $U$ (see inset of Supplementary Fig. 2).

To understand the appearance of the differential conductance asymmetry at the outer steps of the spectra at increased $G_s$, we focus, as an example, on the transition from the ground state which has its main weight in $|m_1, m_2\rangle = |1, \downarrow\rangle$ to the excited state, $|0, \downarrow\rangle$ (solid black arrow in Fig. 3c). During this transition, the spin at the tip stays in the $|\downarrow\rangle$ state while the spin on the sample undergoes a change of $\Delta m_z = -1$ from $|1\rangle$ to $|0\rangle$. This angular momentum has to be provided by the tunnelling electron so that the process only occurs if the electron changes from $|\downarrow\rangle$ to $|\uparrow\rangle$. As Pt is polarized by magnetic impurities such as Co[25–27], we expect the functionalized tip to have an imbalance between spin up and spin down electrons. Assuming

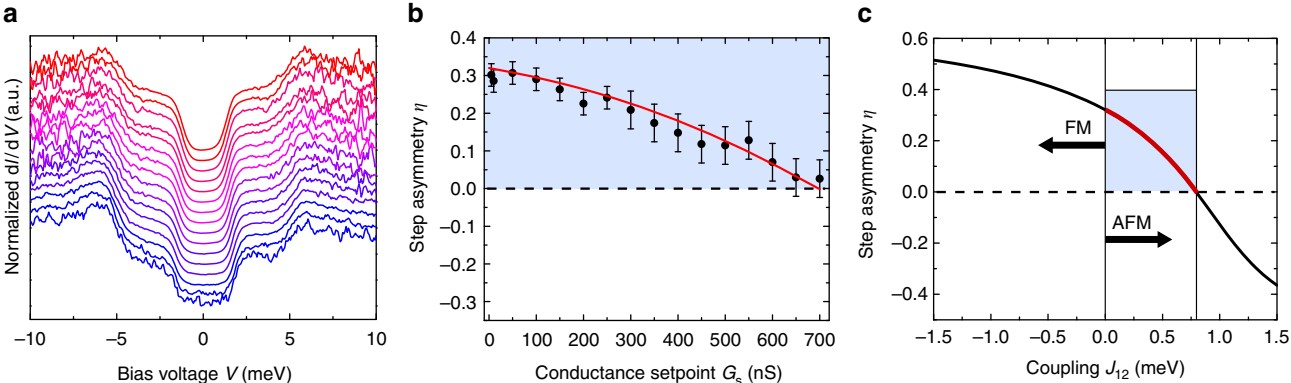

**Figure 4 | Spectroscopic evolution in magnetic field.** (**a**) Evolution of the $dI/dV$ spectra with increasing $G_s$ using a Co-functionalized tip at $B_z = 5\,\text{T}$ (blue $G_s = 20\,\text{nS}$, red: $G_s = 700\,\text{nS}$). A simulation of this spectral evolution is shown in Supplementary Fig. 4. (**b**) Fitting the asymmetry, $\eta_o$, with anisotropy parameters of $D = -4.05\,\text{meV}$, and $E = 0.65\,\text{meV}$, yields an Ising-like antiferromagnetic coupling between both spins of $J_{12} = 1.1\,\mu\text{eV/nS} \times G_s$ and a correlation of $C = 0.6 \pm 0.05$. At $J_{12} = 0.78\,\text{meV}$ $\eta_o$ becomes zero. The error bars include the statistical and systematic error of the fit to the spectroscopic data. In particular, the area under the Gaussian fit to the first derivative of the spectroscopic data in (**a**). (**c**) A simulation that accounts for a wider energy range, shows that only an antiferromagnetic coupling ($J_{12} > 0$) between the spins on tip and sample can reproduce the decrease in $\eta_o$.

an AFM correlation between the state of the tip's spin system and the electrons in the tip, leads to a $|\uparrow\rangle$ polarization, while the weak coupling of the sample spin to the host metal[23] does not lead to any significant polarization. Therefore, for the highlighted transition, the conductance will be enhanced at negative bias, that is, when an electron from the sample reservoir tunnels to the tip reservoir (Fig. 3d). Concomitantly, the conductance is suppressed at positive bias, in agreement with the data presented in Fig. 1g. Importantly, this bias asymmetry is independent of the chosen ground state and a transition from $|-1, \uparrow\rangle$ to $|0, \uparrow\rangle$ with a $|\downarrow\rangle$ tip polarization results in the same observation. An equivalent argument rationalizes the bias asymmetry in Fig. 1h assuming a ferromagnetic (FM) correlation between spin state and electrons in the tip. Note that an explanation based solely on a tip-induced change of the magnetic anisotropy[23,37] can neither account for the asymmetry nor for the decreased intensity of the energetically lower steps (see Supplementary Fig. 3).

**Spin–spin correlations.** We introduce correlations into our transport model by describing the electron bath in the Pt tip by a density matrix, $\hat{\varrho}_2$, which is directly correlated to the spin state of the attached Co atom:

$$\hat{\varrho}_2 = \begin{pmatrix} 0.5 & 0 \\ 0 & 0.5 \end{pmatrix} + C \sum_{i=\text{x,y,z}} \langle \hat{S}_{i,2} \rangle \cdot \hat{\sigma}_i. \quad (2)$$

The correlation strength, $C$, has been fitted to the evolution of $\eta$ with excellent agreement (Fig. 2d,f). We find an AFM correlation, $C = -0.50 \pm 0.05$, for the data set with positive asymmetry (Fig. 1g) and a FM correlation, $C = 0.35 \pm 0.04$, for the set with negative asymmetry (Fig. 1h). To further highlight the validity and quality of our model, we simulate $dI/dV$ spectra by accounting for scattering up to third order in the matrix elements (see 'Methods' section) by considering additional exchange processes between the localized spin on the sample and substrate electrons (Fig. 1i,j)[38]. Note, that this approach considers the localized spins and the bath electrons as separable entities. A full quantum-mechanical description, as for example numerical renormalization group models provide, is beyond the scope of this paper.

**Spectra in magnetic field.** To clarify the sign of the coupling, $J_{12}$, between the spin 1 and spin 1/2, we measure a system similar to that depicted in Fig. 1c, subject to an external magnetic field, $B_z = 5\,\text{T}$ (Fig. 4a,b). For weak coupling (small $G_s$), the spectra show the expected Zeeman-shift of the transition energies and a step asymmetry $\eta_o$ due to field-induced spin-polarization in the tip[31]. With increasing coupling, these two effects are counteracted by the previously described state polarization and correlation effects. At strong coupling, this results in a spectrum that is similar to a bare $S = 1$ spectrum obtained at zero field. In particular, we observe that since $\eta_o$ approaches zero, it is only consistent with AFM coupling, $J_{12} > 0$, between the two spins on tip and sample (see also Supplementary Fig. 5). FM coupling, $J_{12} < 0$, does not fit the data as it would further increase the asymmetry with $G_s$ (Fig. 4c). This measurement, together with the proportionality of $J_{12}$ with $G_s$, allows us to fix the sign of the direct exchange, $J_{12} > 0$, and distinguish between AFM and FM correlations, $C \in [-1,1]$, between the spin on the tip and its electron bath.

## Discussion

In conclusion, we have shown that the correlation between an atomic spin and an electron bath can be determined by coupling it to a second atomic spin in a tunnel junction. The AFM direct exchange coupling between the two atomic spins which can be tuned with $G_s$ is crucial for the determination of the correlation of the strongly hybridized spin with its hosting electron bath. At low $G_s$, when the coupling $J_{12}$ is negligible, we can characterize the unperturbed $S = 1$ CoH spin on the surface. Afterward, at higher $G_s$ and therefore increased $J_{12}$, the sign and strength of $\eta$ makes it possible to distinguish between either AFM or FM correlations between the spin state of the Co adatoms and the electrons in the Pt tip and to quantize its strength. Remarkably, this method enables us to unravel the otherwise hidden spin–spin correlation in these spectroscopically dark and non-distinctive spins. We note, that the different correlation might be due to different Co adatom binding sites on the Pt tip leading to a different coupling mechanism with the substrate, especially on a Pt microfacet of unknown structure[28,39]. In addition, we cannot exclude coupling to other Co atoms in proximity to the apex atom, which could also influence the effective correlation to the tip's electron bath[11,24]. Unexpectedly,

our measurements show that the FM or AFM correlation with the electron bath is related to the direct exchange coupling, which shows either Ising (classical) or Heisenberg (quantum) character. These correlations introduce a measurable transport asymmetry wholly unrelated to static spin polarization and external magnetic fields and might be used as a method to probe correlated electron materials in an inverted tip-sample geometry.

## Methods

**Sample preparation.** The Rh(111) surface was prepared with multiple $Ar^+$ sputtering cycles and annealing up to a temperature of 1,100 K. During the final annealing cycle the temperature was stabilized at 1,080 K and the surface was exposed to borazine ($B_3H_6N_3$) at $1.2 \times 10^6$ mbar for 2 min leading to a self-assembled $h$-BN monolayer. Co atoms were then evaporated onto the sample surface at a temperature of $\approx 20$ K from a Co rod heated by an electron beam. The CoH complexes form during the evaporation from residual hydrogen in the vacuum system.

**Spectroscopy.** Spectroscopy ($dI/dV$) was measured using an external lock-in amplifier and modulating the bias voltage with a sinusoidal of 0.2 mV amplitude and a frequency of 689 Hz. The conductance setpoint of the tunnel junction ($G_s = I_s/V_s$) is defined by the applied bias voltage to the sample, $V_s$, and the setpoint current, $I_s$. This conductance setpoint defines the distance between tip and sample and also the coupling strength $J_{12}$. We disable the $I_s$ feedback loop in order to take the $dI/dV$ spectrum at a constant distance between tip and sample. For measurements in magnetic field, an external field of 5 T was applied along the surface normal. All experiments were performed in ultrahigh vacuum ($\approx 10^{-10}$ mbar) and a base temperature of 1.4 K.

**Tip functionalization.** Bare Pt tips from 25 μm wire have been functionalized by positioning the tip above a CoH complex at a setpoint of $I_s = 20$ pA and $V_s = -15$ mV. From this setpoint, we decrease the tip-sample separation until a jump in the current is observed. The surface area is then scanned to confirm vertical atom manipulation. We assume that the hydrogen detaches from the CoH during this manipulation due to the absence of spectroscopic features in our bias range[29] (see Supplementary Fig. 1). Successful preparation of Co-functionalized tips results in a sharper topographic contrast[31].

**Simulations.** To simulate the electron transport and calculate the $dI/dV$ spectra, we adapt a perturbative scattering model in which spin-flip processes up to the second order Born approximation are accounted for and which has been previously successfully used on different quantum spin systems[9,16,23,38]. In this model, the two electron reservoirs of tip and sample are described by

$$H_j = \sum_{\substack{j=1,2 \\ k,\sigma}} \epsilon_{jk\sigma} \hat{a}_{jk\sigma}^\dagger \hat{a}_{jk\sigma}, \tag{3}$$

with $\hat{a}^\dagger(\hat{a})$ as the creation (annihilation) operators in second quantization for electrons at the electrode $j=1$ (sample) and $j=2$ (tip) with momentum $k$, spin $\sigma$ and the energy $\epsilon_{jk\sigma}$. The total Hamiltonian writes than as

$$H = H_0 + H_1 + H_2 + H', \tag{4}$$

with $H'$ describing the tunnelling of electrons from tip to sample or vice versa via Kondo-like spin-flip or potential scattering processes and, additionally, the scattering of the bath electrons with the impurity:

$$H' = V_{1\to 2} + V_{2\to 1} + V_{1\to 1} + V_{2\to 2},$$
$$V_{j\to j'} = \sum_{\sigma,\sigma'} T_0 \hat{a}_{j'\sigma'}^\dagger \hat{a}_{j\sigma} \left( \frac{1}{2}\hat{\boldsymbol{\sigma}} \cdot \hat{\mathbf{S}} + U\hat{\sigma}_0 \right),$$
$$V_{j\to j} = \frac{1}{2} \sum_{\sigma,\sigma'} J_j \hat{a}_{j\sigma'}^\dagger \hat{a}_{j\sigma} \hat{\boldsymbol{\sigma}} \cdot \hat{\mathbf{S}}. \tag{5}$$

Transition rates between the initial $|\psi_i\rangle$ and final $|\psi_f\rangle$ eigenstates of $H_0$ due to the interaction with electrons originating from the reservoir $j$ and absorbed in $j'$ are calculated using Fermi's golden rule:

$$\Gamma_{if}^{j\to j'}(eV) = \frac{G_{jj'}}{e^2} \int_{-\infty}^{\infty} d\epsilon\, W_{i\to f} f(\epsilon - eV, T)[1 - f(\epsilon - \epsilon_f + \epsilon_i, T)], \tag{6}$$

with $f(\epsilon, T)$ as the Fermi-Dirac distribution, $T = 2$ K the effective temperature in our experiment, $G_{12} = G_{21} = T_0^2 e^2$ adjusted to match the experimentally set $G_s$ and $W_{i\to f}$ the transition probabilities evaluated up to second order Born approximation:

$$W_{i\to f} = \frac{2\pi}{\hbar} \left( |M_{i\to f}|^2 + J_1\rho_1 \sum_m \left( \frac{M_{i\to m}M_{m\to f}M_{f\to i}}{\epsilon_i - \epsilon_m} + \text{c. c.} \right) \right). \tag{7}$$

Approximating the electron baths in tip and sample by the energy

independent spin density matrices, $\rho_j$, the Kondo-like scattering matrix elements (neglecting potential scattering) can be written as $M_{i\to j} = \sum_{i',j'} \sqrt{\lambda_{i'}\lambda_{j'}} \langle \boldsymbol{\sigma}_{j'}, \psi_j | \hat{\sigma} \cdot \hat{\mathbf{S}} | \boldsymbol{\sigma}_{i'}, \psi_i \rangle$. The $\boldsymbol{\sigma}_{i',j'}$ are the eigenvectors and $\lambda_{i',j'}$ the eigenvalues of the density matrices $\varrho$ of the electrons in tip and sample participating in the scattering process, which are influenced by the correlation between the localized spins and the electrodes (equation 2)[38]. The first term in equation (7) is responsible for the conductance steps observed in the spectra, while the second term leads to logarithmic peaks at the intermediate energy $\epsilon_m$ and scales with the dimensionless coupling $J_1\rho_1$ between the sample electrons and the CoH spin with $J_1$ as the coupling strength and $\rho_1$ as the density of states in the sample close to the Fermi energy[38]. For the systems discussed in this paper 1, we found $J_1\rho_1 \approx -0.05 - 0.1$.

The set of rate equations (equation 6) enables us now to build characteristic master equations for the state populations $p_i$ in which we take excitations and de-excitations of the spin system by the tunnelling electrons and bath electrons into account[24,38]:

$$\frac{d}{dt}p_f = \sum_{i\neq f} \sum_{j,j'} p_i \Gamma_{if}^{j\to j'} - p_f \sum_{j\neq i} \sum_{j,j'} \Gamma_{fi}^{j\to j'}. \tag{8}$$

Note that for the transport between tip and sample, we only account for scattering on the spectroscopically active $S_1$. From the steady-state occupation $p_i^{stat}(eV)$ and the rates we can continue to calculate the current

$$I(eV) = e \sum_{i,j} p_i^{stat}(eV) \left( \Gamma_{ij}^{2\to 1}(eV) - \Gamma_{ij}^{1\to 2}(eV) \right), \tag{9}$$

and by numerical differentiation $dI/dV$. Spin-pumping effects are strongly damped due to an effective interaction of the sample conduction electrons with the CoH spin, which we found to be in the order of $G_{11} \approx 2$ μS. While similar values have been found for other atomic spin systems[31,34], we mark that $G_{11}$ is higher than the $\frac{2e^2}{\hbar}(J_1\rho_1)^2 \approx 0.2 - 0.8$μS from the spectroscopically visible coupling[33,38]. However, the influence of $G_{11}$ on $\eta_o$ and $\eta_i$ is only small (see Supplementary Fig. 5). We note that better theoretical models can be developed in order to further understand the behaviour of the materials.

**Data availability.** The relevant spectroscopic data sets used in this publication are available from the authors.

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

## Acknowledgements

We thank Oleg Brovko, Lihui Zhou, Sebastian Loth, Maciej Misiorny, Philipp Hansmann and Fabian Pauly for fruitful discussions as well as Gennadii Laskin for his help with the experiment. P.J. acknowledges support from the Alexander von Humboldt Foundation. M.M. and M.T. acknowledge support from the SFB 767.

## Author contributions

M.T. and K.K. conceived the experiments. M.M. and P.J. performed the STM measurements. M.M. analysed and fitted the data. M.T. developed the transport model. All authors discussed the results and contributed to the manuscript.

## Additional information

**Competing financial interests:** The authors declare no competing financial interests.

**How to cite this article**: Muenks, M. *et al.* Correlation-driven transport asymmetries through coupled spins in a tunnel junction. *Nat. Commun.* **8,** 14119 doi: 10.1038/ncomms14119 (2017).

**Publisher's note**: 

