## [Peer Review File · Nature Communications]

Reviewers' comments:

Reviewer #1 (Remarks to the Author):

The authors investigate the transport properties of a nanojunction formed by a Co-functionalized platinum tip and a CoH complex decoupled from the underlying Rh substrate by an insulating h-BN monolayer. By varying the conductance set-point G_s , the differential conductance dI/dV shows an increasing asymmetry in the height of the outermost cotunneling steps. Whether the step height is larger at negative or positive bias voltages, is sample dependent. In contrast, in the case in which the Pt tip is not functionalized no asymmetry is found by variation of the conductance set-point.

To explain this difference, the authors suggest that the Co atom on the tip interacts antiferromagnetically or ferromagnetically with the surrounding tip electrons, thus polarizing locally the platinum electrons, and in turn favoring specific inelastic processes from the sample CoH complex to the tip or vice-versa.

It is concluded that a higher step at positive/negative bias is a signature of antiferromagnetic/ferromagnetic correlations.

Finally, an effective Kondo model is used to describe these correlations and which give transport spectra in good agreement with the experiment.

Regarding the story line, the authors first analyze the spectrum of the CoH alone and then that of the coupled dimer as a function of the conductance set-point. For both cases a model Hamiltonian is proposed. While I find the model regarding the simple and standard case of a CoH complex alone convincing, I am neither persuaded nor satisfied with the part regarding the functionalized tip. In particular, I am not convinced by the fact that the coupling assumed in Eq. (1) can have a Heisenberg or Ising form depending on the sample. Similarly, I am not persuaded by the correlation term in Eq. (2) which has ferromagnetic or antiferromagnetic character depending on the considered sample.

Asymmetric inelastic cotunneling spectra are common in the literature concerning multilevel quantum dots. They simply require an asymmetry between source and drain, which is very likely the case in the STM set-up investigated in this work, and different tunneling couplings to the different levels.

Importantly, Kondo correlations will enhance the asymmetry, but are not the reason for it, as shown e.g. in the work by Paaske et al., Nature Physics 12, 460 (2006).

To my understanding, what can be clearly evinced from the experiment, is that the presence of the Co atom on the tip affects the magnetic anisotropies E and D of the CoH complex, which acquire an explicit dependence on G_s . Likewise for the tunneling couplings.

What I miss here is a convincing minimal model Hamiltonian accounting for the complexity of a spin $S=1$ junction tunnel coupled to a Co-functionalized tip and to the Rh-substrate capable of describing in a unique framework the behavior shown in Figs. 2b, 2c, 2e and 2f.

Thus, I cannot recommend publication of this work in Nature Communications unless profound revisions of the manuscript are undertaken.

Reviewer #2 (Remarks to the Author):

The work by Muenks et al is a very interesting piece of research showing unforeseen aspects of electron tunneling when spin variables need to be considered. It is an excellent research piece with careful and extraordinary experimental data, revealing spin excitations in a complex system (two Co atoms coupled to electrodes). But in my opinion, what really makes this work unique is the analysis

that explains the data. I think that this work will create a current of opinion and development after publication. For this reason I recommend its publication.

I would like to have the authors consider the following points with the aim of clarifying/improving the exposition of their findings.

1. I think the work could improve by reducing the number of figures (or panels) and increasing the explanation. Figures are powerful because they can quickly convey an idea, but they are also difficult to understand because they can contain a lot of information. Particularly, I am not sure that one needs figure 1, because the text is clear and the experiment is mainly tunneling, which is difficult to sketch in a graph. Figure 3 (b) and (c) should clearly show that they correspond to different spin-spin interactions (it took me a while to realize that and it is an important piece of information). Figure 3 (g) and (h) are a bit complex to me. I would suggest to use the space left by the removal (or perhaps simplification) of figure 1 and make a full figure for them. 3D graphs contain a lot of information and are difficult to grasp (I acknowledge I am particularly lazy with this type of graphs...)

2. I think the explanation of the data and the new modeling are probably correct. But I was wondering if we could rule out spin pumping. I am afraid this is not discussed by the authors in the text, and only one obscure sentence in the conclusions is found ("wholly unrelated to static spin polarization"). Of course, spin-polarized electrons are needed for spin pumping, but I was wondering if the Co-Co interaction would not offer this: the tip's Co atom becomes polarized by the substrate's Co atom. Have the authors considered this? Have they simulated it? When they apply a magnetic field (Figure 4) this is definitely a possibility.

3. The paper emphasizes the "correlation" aspects of the physics revealed in this work. I am a bit mystified by the extensive use of the term "correlation" in abstract and introduction, and then somehow its use is damped out in the rest of the paper. I think the authors really show that "correlation" is at play, but they should state what kind of correlation. I initially thought it was the correlation caused by Heisenberg interactions between spins, but the authors show that mainly Ising interactions suffice, which have no correlation. I then understood that the authors refer to the electron-spin interaction that brings in Kondo correlations. I think this should be clearly stated both at the beginning and in the modeling. In the same way, some statements should be toned down because the model ling is only partially taking care of the correlations.

4. In the same line, I would enjoy a longer exposition of the model ling. I think I understand what the authors have done, but I would like to fully understand the underlying assumptions, for example, different degrees of freedom are decoupled and the spin degrees of freedom are treated to third order in the substrate's impurity spin, but the tip's spin is sort of averaged. The paper would improve from some more details.

In Summary, this is an excellent and original piece of research that adds a new twist to the field of dynamical processes of spins on the nanoscale. I fully recommend publication in Nature Communications.

Reviewer #3 (Remarks to the Author):

The paper aims at studying the transport asymmetry through a tunnel junction made of coupled spins in an STM junction. For this purpose the authors reproduce the configuration of a CoHx on a boron nitride surface of their earlier work (Ref 23). The study also exploits the spin-polarized STM experiment described by Loth and co-worker (Ref 24). It is supposed to inform on correlations

between a spin impurity (CoHx) and its electron bath. The quality of the data/presentation makes it an interesting piece of work.

To analyse the the asymmetry of conductance of the two spin impurity STM junction the authors fit the spectroscopic data with the same spin Hamiltonian as in ref. 23 to which a direct exchange term J between sample and tip is added. It is found that the direct exchange coupling J is AFM and proportional to the conductance and decreases exponentially with distance between the two spins. The work described in this paper is based on a series of intricate propositions. In particular, the notion of correlation is used with a different meaning in different places throughout the manuscript. It seems to me that the authors should clarify this issue. The position adopted in the work calls for a series of questions and remarks:

1) Magnetic moments are inferred (tip) or obtained from the fitting (sample). The transport is again described, by a Kondo type interaction (Ref 23). Correlation is intrinsic to this kind of experiment, therefore, to make life easier for the reader, the authors should make clear from the beginning what is the novelty compared to previous work, i.e. ref 22 and 23.

2) According to the abstract, the work is supposed to give information on the correlation between a spin impurity and the electron bath (CoH). However the interpretation necessitates treating the correlation within the tip itself (P:6) and between the tip and the sample. It would be useful to know what is the input what is the output?

3) In the "Method section" it is assume that hydrogen detaches from the CoH during vertical manipulation. This assumption should be justified since the dissociation energy is about 2 eV, much larger than the ones used for the manipulation. This point has its importance for the choice of the Co terminated Pt tip. Two different sets of measurements are shown in Fig. 3, in both cases it is assumed that the tip is made of Co on Pt. Is there any particular reason for this choice?

4) Changing the spin states of an impurity on a surface by tuning the current of functionalized tips has been used before. It would be interesting if the authors could discuss what is the link between the spin torque experiment (Loth) with the same configuration and their own work where a direct exchange between tip and sample impurity is assumed?

Point-by-Point response to the referees' comments

Questions and comments raised by referee #1:

1.1. Referee #1 writes when he/she summarizes our paper: “...Whether the step height is larger at negative or positive bias voltages, is sample dependent.”

Here our manuscript was evidently not clear enough. Sample refers to the CoH $S=1$ complex adsorbed on the h -BN/Rh(111) substrate. We found that the difference in the asymmetry is sample independent in the experiment and the used model. It is the tip with its attached Co atom which makes the difference. This is now explicitly stated in the text, see additions below.

1.2. Referee #1 continues that he/she is “... not convinced by the fact that the coupling assumed in Eq. (1) can have a Heisenberg or Ising form depending on the sample. Similarly, I am not persuaded by the correlation term in Eq. (2) which has ferromagnetic or antiferromagnetic character depending on the considered sample.”

Also here our manuscript was apparently not clear enough that led to a misunderstanding of an important point of our work, that is the role of tip and sample: We find the different couplings and correlations depending solely on the tip, but independent of the particular CoH $S=1$ system on the sample. We find Heisenberg-like coupling between the spin attached to the tip apex and CoH spin on the sample when the tip's spin is AFM correlated with the tip's bath electrons. Additionally, we find Ising-like coupling between the spin of the tip and the one of the sample when the tip's spin is FM correlated with the tip's bath electrons.

However, we don't know the exact cause of this different behavior. In our manuscript we base our assumption on the energy evolution of the inner and outer steps in the spectra when varying the setpoint G_s , i.e. the distance z between the spins on sample and tip. We note, that we do discuss this issue in the main text:

“We note, that the different correlation might be due to different Co adatom binding sites on the Pt tip leading to a different coupling mechanism with the substrate, especially on a Pt microfacet of unknown structure. Additionally, we cannot exclude coupling to other Co atoms in proximity to the apex atom which could also influence the effective correlation to the tip's electron bath.”

To make the difference between tip and sample more clear we have changed the text of the manuscript on several points. In particular in the abstract we write now:

“...Here, we determine the correlations between a strongly hybridized spin impurity on the tip of a scanning tunneling microscope...” and later “...this asymmetry can be controlled by the coupling strength and is directly related to either ferromagnetic (FM) or antiferromagnetic (AFM) spin-spin correlations in the tip.”

Furthermore we updated the figures to make it easier to understand that it is the tip which is different for the 3 datasets discussed throughout the manuscript (see also answer to referee #2).

1.3. Referee #1 writes: “Asymmetric inelastic cotunneling spectra are common in the literature concerning multilevel quantum dots. They simply require an asymmetry between source and drain, which is very likely the case in the STM set-up investigated in this work, and different tunneling

couplings to the different levels. Importantly, Kondo correlations will enhance the asymmetry, but are not the reason for it, as shown e.g. in the work by Paaske et al., Nature Physics 12, 460 (2006).“

The referee is fully correct, that the STM junctions discussed in our manuscript are inherently asymmetric. The coupling of the two spins to their corresponding electrodes (tip or sample) is much stronger than the coupling between tip and sample. However, the Coulomb charging energy in quantum dots (as in the work by Paaske et al.) is relatively small due to their much larger spatial extension compared to the energy in our well defined atomic spin systems. Their charging energy is in the eV range, far outside the energy range of all observed excitations.

Thus, we are in the deep Coulomb-blockade regime where an effective co-tunneling Hamiltonian which consists of a Kondo-like and an additional potential scattering term can be found using a Schrieffer-Wolf transformation (see for example Ref. 37 where such an approach is used). The potential scattering term enables us to describe the different co-tunnel channels in which either an additional electron or hole is virtually occupying the spin site. In our model, the potential scattering is a parameter which we safely neglect.

However, we believe that the question raised by the referee is important. While the outer step asymmetry is only negligibly influenced, the inner step asymmetry, which we have now added to figure 2, is slightly influenced by a potential scattering term (see also the insets of Supplementary Figure 2) and might even be the reason for the small systematic offset of the inner step asymmetry.

Therefore we have updated the manuscript and write:

“We now continue to describe the electrical transport through the junction by employing an interaction Hamiltonian, $\frac{1}{2} \sigma \cdot S + U$, between the tunneling electrons and the coupled spin system, with $\sigma = (\sigma_x, \sigma_y, \sigma_z)$ as the standard Pauli matrices and $S = S_1 \otimes S_2$ as the combined spin operator of the two spins. The first term of this interaction describes Kondo-like spin flip scattering processes, while the second term describes spin-conserving potential scattering processes^{25, 37}. The potential scattering processes have only marginal influence on the spectrum and, in particular, on the asymmetry η_o (see Supplementary Fig. 2). Therefore we will neglect it in the following by setting $U = 0$. The systematic offset between observed and calculated η_i could be due to a non-zero U (see inset of Supplementary Fig. 2).”

Furthermore, we now provide a Supplemental Figure 2 explicitly showing the influence of an additional potential scattering term U on the spectra.

1.4. Referee #1 writes: “To my understanding, what can be clearly evinced from the experiment, is that the presence of the Co atom on the tip affects the magnetic anisotropies E and D of the CoH complex, which acquire an explicit dependence on G_s . Likewise for the tunneling couplings.”

We are very happy that the referee asks this interesting possibility. To address this point we now show in the supplemental material a simulation in which we only assume that the magnetic anisotropies change with the conductance G_s in a way to match the experimentally observed step energy. From this graph it can be clearly seen that such a model is neither describing the observed asymmetry nor the reduction in transition intensity of the inner steps. We also add to the manuscript the following sentence:

“Note that an explanation based solely on a tip induced change of the magnetic anisotropy^{24, 33} can neither account for the asymmetry nor for the decreased intensity of the energetically lower steps (see Supplementary Fig. 3).”

1.5. Referee #1 writes: “What I miss here is a convincing minimal model Hamiltonian accounting for the complexity of a spin $S=1$ junction tunnel coupled to a Co-functionalized tip and to the Rh-substrate capable of describing in a unique framework the behavior shown in Figs. 2b, 2c, 2e and 2f.”

We agree with the referee that our model was not well explained. Now we use the additional space the Nature Communication format provides us to **outline our model used for the calculation in much more detail in the Method section**. We hope that this still brief description satisfies the referee as well as the future reader of our manuscript. However, we would like to remark that this paper is mainly about an experimental research and cannot provide a full-fledged theoretical model. Here, we use a simple model which enables us to describe the main features of our experimental observation. We hope, however, that our experimental work will stimulate theoreticians to apply more advanced methods for describing our data.

Questions and comments raised by referee #2:

2.1. Referee #2 proposes an improvement of the manuscript: “...by reducing the number of figures (or panels) and increasing the explanation. Figures are powerful because they can quickly convey an idea, but they are also difficult to understand because they can contain a lot of information. Particularly, I am not sure that one needs figure 1, because the text is clear and the experiment is mainly tunneling, which is difficult to sketch in a graph. Figure 3 (b) and (c) should clearly show that they correspond to different spin-spin interactions (it took me a while to realize that and it is an important piece of information). Figure 3 (g) and are a bit complex to me. I would suggest to use the space left by the removal (or perhaps simplification) of figure 1 and make a full figure for them. 3D graphs contain a lot of information and are difficult to grasp (I acknowledge I am particularly lazy with this type of graphs...”

We gladly follow the advice of the referee and have updated figures and panels to make our story more accessible to the reader. The figures have now the following structure:

Fig. 1 is now a merged version of the old figure 1 and 2. To make it more clear that the panels show results from three different datasets we have put the schematic images of the old figure 1 above the experimentally obtained results and mark clearly that they are measured with different tips.

Fig. 2 contains now the panels (a)-(f) of the old figure 3. To make the relation to the data shown in figure 2, we add labels above the panels and mark the sign of the correlation C . Additionally, we now also show in panel (e) and (f) the experimentally obtained as well as the simulated asymmetries of the inner steps.

Fig. 3 contains now the old panels (g), (h), and (i). To make the 3D graph better accessible for the reader we add additionally a 2D plot at zero coupling. Furthermore, we unify the style of the 2D plots with the one in Figure 1h.

Fig. 4 stays unaltered.

2.2. Referee #2 asks questions concerning our model and alternative explanations. He/she writes: “...I was wondering if we could rule out spin pumping. I am afraid this is not discussed by the authors in the text, and only one obscure sentence in the conclusions is found (“wholly unrelated to static spin polarization”). Of course, spin-polarized electrons are needed for spin pumping, but I was wondering if the Co-Co interaction would not offer this: the tip's Co atom becomes polarized by the substrate's Co

atom. Have the authors considered this? Have they simulated it? When they apply a magnetic field (Figure 4) this is definitely a possibility.”

We fully agree with the referee that the explanation of our model was too brief. We have changed this and **outline now our model used for the calculation in much more detail in the Method section**. Here we note that our simulations are not calculated in the zero-current approximation, but using rate equations and therefore also cover spin-pumping effects.

We now add to the manuscript **a supplemental figure, which shows the influence of different couplings with the sample on the obtained spectra (Supplementary Figure 5)**. With them we can clearly rule out that spin-pumping effects are the main reason for the observed asymmetry.

However, the referee is correct that at high G_s the Co-Co AFM coupling lead to an AFM correlation between the two spins which is needed for the detection of the asymmetry. This is illustrated **in figure 3** and discussed in the manuscript under “Discussion” where we write:

“... The consistently found AFM direct exchange coupling between the two atomic spins which can be tuned with G_s is crucial for the determination of the correlation of the strongly hybridized spin with its hosting electron bath. At low G_s , when the coupling J_{12} is negligible, we can characterize the unperturbed $S = 1$ CoH spin on the surface. Afterward, at higher G_s and therefore increased J_{12} , the sign and strength of η makes it possible to distinguish between either AFM or FM correlations between the spin state of the Co adatoms and the electrons in the Pt tip and to quantize its strength.”

2.3. Referee #2 writes: “The paper emphasizes the “correlation” aspects of the physics revealed in this work. I am a bit mystified by the extensive use of the term “correlation” in abstract and introduction, and then somehow its use is damped out in the rest of the paper. I think the authors really show that “correlation” is at play, but they should state what kind of correlation. I initially thought it was the correlation caused by Heisenberg interactions between spins, but the authors show that mainly Ising interactions suffice, which have no correlation. I then understood that the authors refer to the electron-spin interaction that brings in Kondo correlations. I think this should be clearly stated both at the beginning and in the modeling. In the same way, some statements should be toned down because the model ling is only partially taking care of the correlations.”

We agree with the referee that, on some points, we were not precise enough with the use of the word “correlation”. We hope that our updated version is better in describing the correlations at play. The Nature Communication format enabled us to add a Paragraph labeled **“Spin-spin correlation”**. Additionally, we clarify throughout the manuscript that the correlation C is the spin-spin correlation between the Co adatom on the tip and the tip's electron bath.

The referee is also right, that there is more than one correlation at work. As discussed in answer 2.2. we explain this detailed in our “Discussion” section.

In this context we would also like to note, that Ising interactions between the spin on the sample and on the tip lead to (classical) correlations which are indeed sufficient for the observed asymmetries.

The Kondo-like scattering between the $S=1$ spin of the CoH on the sample and the electrons therein, described by the coupling parameter $J_1\rho_1$ and studied in detail in our earlier publication (Ref. 23) does not produce any significant spin-spin correlation in the sample substrate. In Ref. 23 we found that this coupling, due to the h-BN decoupling layer, is relatively weak and can lead to an energy renormalization of the magnetic anisotropy parameters D and E . In the experiment discussed here these parameters are static for the CoH $S=1$ complex and are not influenced or changed by the approaching bare tip (compare figure 1e main text). In fact, for the Co-functionalized tips at low coupling strengths

(large separation distances z and therefore $J_{12} \sim 0$) we observe the spectrum of an unperturbed CoH $S=1$ system.

Concerning the modeling we have toned down our approach and write now:

“...Note, however, that this approach considers the localized spins and the bath electrons as separable entities. A full quantum-mechanical description, as for example numerical renormalization group models would provide, is beyond the scope of this paper...”

2.4. Referee #2 writes: “In the same line, I would enjoy a longer exposition of the modelling. I think I understand what the authors have done, but I would like to fully understand the underlying assumptions, for example, different degrees of freedom are decoupled and the spin degrees of freedom are treated to third order in the substrate's impurity spin, but the tip's spin is sort of averaged. The paper would improve from some more details.”

We now provide details about the model in the method section of the manuscript.

Questions and comments raised by referee (3):

3.1 Referee #3 writes in his/her summary of our manuscript: “The work described in this paper is based on a series of intricate propositions. In particular, the notion of correlation is used with a different meaning in different places throughout the manuscript. It seems to me that the authors should clarify this issue.”

Referee #2 raised a similar question (2.3.) and we would like to point to our answer there. In particular, we would like to point again to our much more detailed and specified “Discussion” section.

3.2. Referee #3 writes: “Magnetic moments are inferred (tip) or obtained from the fitting (sample).”

Here, we would like to note that for the CoH spin on the sample we know precisely its spin $S=1$ (Ref. 23). Only the exact values of the magnetic anisotropy and the coupling to the sample electrons are determined from the spectrum at low G_s . We approximate the tip spin as the simplest possible spin $S=1/2$ but note in Ref. 34 that a total spin of $S=3/2$ would equal well fit the data.

3.3 Referee #3 asks: “The transport is again described, by a Kondo type interaction (Ref 23). Correlation is intrinsic to this kind of experiment, therefore, to make life easier for the reader, the authors should make clear from the beginning what is the novelty compared to previous work, i.e. ref 22 and 23. According to the abstract, the work is supposed to give information on the correlation between a spin impurity and the electron bath (CoH). However the interpretation necessitates treating the correlation within the tip itself (P:6) and between the tip and the sample. It would be useful to know what is the input what is the output?”

The referee is right, that correlations are intrinsic in such experiments, however, it's quantification is difficult. Similar as for referee #1, also for referee #3 the manuscript was not clear enough to explain which correlation is determined in our experiment. We hope that our modified abstract and updated manuscript is now addressing this point more clearly:

“Here, we determine the correlations between a strongly hybridized spin impurity on the tip of a

scanning tunneling microscope and its electron bath by varying the coupling to a second spin impurity weakly hybridized to the sample surface.”

3.3. Referee #3 writes: “In the "Method section" it is assume that hydrogen detaches from the CoH during vertical manipulation. This assumption should be justified since the dissociation energy is about 2 eV, much larger than the ones used for the manipulation. This point has its importance for the choice of the Co terminated Pt tip. Two different sets of measurements are shown in Fig. 3, in both cases it is assumed that the tip is made of Co on Pt. Is there any particular reason for this choice?”

We thank the reviewer for pointing out that we did not address the hydrogen detachment process during the vertical manipulation. While the details of this process are unknown to us, we know whether the hydrogen is still attached to the Co from spectroscopic measurements.

For this we now show in the Supplementary Figure 1 spectroscopic data of bare Pt tips and Co as well as CoH functionalized Pt tips. Pt tips with an attached CoH complex have a distinctive dip around zero voltage, contrary to our used Co/Pt tips. This is in accordance to previously reported work on CoH complexes on Pt(111) (see Ref. 29).

Therefore, we now write in the manuscript:

“...In our energy range of interest bare Pt as well as Co-functionalized tips are spectroscopically nondescriptive (see Supplementary Fig. 1).”

However, the comparison with a Pt(111) surface might be still different from a multifacet Pt tip apex. Note, however, that we discuss this issue in our manuscript where we write in the “Discussion” section:

“Here we note, that different Co adatom binding sites on the Pt tip can lead to a different coupling mechanism with the substrate, especially on a Pt microfacet of unknown structure. Additionally, we cannot exclude coupling to other Co atoms in proximity to the apex atom which could also influence the effective correlation to the tip’s electron bath”

3.4. Referee #3 writes: “Changing the spin states of an impurity on a surface by tuning the current of functionalized tips has been used before. It would be interesting if the authors could discuss what is the link between the spin torque experiment (Loth) with the same configuration and their own work where a direct exchange between tip and sample impurity is assumed?”

We believe the referee is pointing with this question to the reference 24 (Loth, et al. Nat. Phys. 6, 340 (2010)). In this experiment, Mn atoms on CuN were probed with a functionalized tip. Crucial for that experiment was the external magnetic field which permanently spin-polarized the tip. This spin-polarized current exerted a spin torque on the Mn atom. However, in our experiment, without an externally applied magnetic field, the time-average of our polarization is zero and can therefore not lead to any spin torque.

To clarify this point we now write in the manuscript:

“Here, an exchange coupling of $J_{12} = 2$ meV is sufficient to polarize the ground and first excited states in the doublets with weights greater than 0.85 (Fig. 3c). However, as long as no external magnetic field breaks the degeneracy of the doublets the time-average magnetization of the spin-system stays always zero.”

Nevertheless, for the data measured in an external magnetic field of $B = 5$ T shown in Fig.4, spin torque effects do exist. We don't discuss these effects because they are not the focus of the paper.

REVIEWERS' COMMENTS:

Reviewer #1 (Remarks to the Author):

In the revised version of the manuscript the authors have tried to improve the readability of their work and to make more clear which kind of "correlations" they are addressing. Those are antiferromagnetic (AFM) or ferromagnetic (F) correlations between the Pt electrons on the tip and a Co atom hybridized with it.

Those correlations are parametrized by a constant C being positive or negative for AFM/F correlations, respectively. To support their claim, transport calculations are provided which can reproduce the experiment.

I am not satisfied with the present version of the manuscript, of which I cannot recommend its publication. Specifically:

i) That at low enough energies spin-spin correlations should exist between the conduction electrons on the tip and the spin impurity is a trivial statement.

What is non trivial is to understand why the correlations are of AFM or F type and to corroborate this statement with a microscopic model. Here the authors find both signs of C , and do not have any way to predict which experimental configuration yields AFM or FM coupling. A microscopic derivation of the correlation mechanism yielding to positive or negative C is not presented.

ii) The authors present cotunneling transport calculations which include Kondo-like terms in the total Hamiltonian. Unfortunately, the new part included in the method section is very confused and with several mistakes or imprecisions. In general I find the theoretical treatment unclear and not at the level of a Nature Communication paper.

In particular:

While Eqs. (6)-(9) are standard, their relation to Eqs. (2) -(5) is fully unclear.

For example, it is unclear how starting from the full Hamiltonian Eq. (4) the leads degrees of freedom (cf. Eq. (3)) with the Kondo couplings (5) are traced out in order to get the effective rates (6) and (7). Note also that the definition of the total spin S is wrong; the indices as well as the mathematical definition of the coupling in Eq. (5) is imprecise; there is no clear relation between parameters like T_0 and G_{jj} .

Reviewer #2 (Remarks to the Author):

The authors have satisfactorily responded to each and every comment by the three referees. Thus, I recommend publication in its present form.

Reviewer #4 (Remarks to the Author):

The subject of the paper is important and timely. The points raised by the referees were well answered and the paper is much clearer now. So, for me, it's perfect for publication. Felicitations to the authors for their high-quality experiments.

Point-by-Point response to the referees' comments

Questions and comments raised by referee #1:

1.1 Referee #1 writes: “That at low enough energies spin-spin correlations should exist between the conduction electrons on the tip and the spin impurity is a trivial statement.”

While the appearance of some spin-spin correlations between an impurity spin and the interacting host electrons might be a trivial statement, the quantification of their sign and strength is certainly not trivial. In our manuscript, we show how a STM can be utilized to measure these correlations.

1.2 Referee #1 writes: “What is non trivial is to understand why the correlations are of AFM or F type and to corroborate this statement with a microscopic model. Here the authors find both signs of C, and do not have any way to predict which experimental configuration yields AFM or FM coupling. A microscopic derivation of the correlation mechanism yielding to positive or negative C is not presented.”

We fully agree with the referee that an understanding of the microscopic origin of the different types and strengths of the correlations in the tip would be admirable. However, this is not the aim of the manuscript and is very difficult to address without speculation. The principal difficulty, that the tip is less well defined than the sample, is inherent to almost all STM measurements.

Nevertheless, we provide possible reasons for the observed behavior and place our results in the correct context by citing the relevant literature. We note this in the manuscript as, “...We note, that the different correlation might be due to different Co adatom binding sites on the Pt tip leading to a different coupling mechanism with the substrate, especially on a Pt microfacet of unknown structure. Additionally, we cannot exclude coupling to other Co atoms in proximity to the apex atom which could also influence the effective correlation to the tip’s electron bath....” (lines 176-180)

Furthermore, we outline possible future extensions by writing: “...These correlations introduce a measurable transport asymmetry wholly unrelated to static spin polarization and external magnetic fields and might be used as a method to probe correlated electron materials in an inverted tip-sample geometry.” (lines 182-184)

1.3. Referee #1 writes: “The authors present cotunneling transport calculations which include Kondo-like terms in the total Hamiltonian. Unfortunately, the new part included in the method section is very confused and with several mistakes or imprecisions. In general I find the theoretical treatment unclear and not at the level of a Nature Communication paper.”

We thank the referee for pointing out some formatting inaccuracies in our model. In the revised version we have corrected the **indices in equ. (5)**. Furthermore, we have **removed the approximate in equ. (7)** by adding the correct prefactor. We believe that these small changes makes the model better accessible.

To further point out that more advanced theoretical models might clarify the behavior in our experiment, we now add to the end of the method section the sentence: “**We note that better theoretical models can be developed in order to further understand the behavior of the materials**”. (lines 244-245)

1.4. Referee #1 writes: “... there is no clear relation between parameters like T_0 and G_{jj} .”

We apologize for this missing point and state now: “... $G_{12} = G_{21} = T_0^2 e^2$...” (line 221)

1.5 Referee #1 writes: “While Eqs. (6)-(9) are standard, their relation to Eqs. (2)-(5) is fully unclear. For example, it is unclear how starting from the full Hamiltonian Eq. (4) the leads degrees of freedom (cf. Eq. (3)) with the Kondo couplings (5) are traced out in order to get the effective rates (6) and (7).”

To solve the rate equations, it is necessary to calculate the different matrix elements M of equation (6). To do so we approximate the electron baths in tip and sample by the energy independent spin density matrices which are modified accordingly to equation (2). To clarify more that the spin density matrices are modified we now write: “The $\sigma_{i,j}$ are the eigenvectors and $\lambda_{i,j}$ the eigenvalues of the density matrices of the electrons in tip and sample participating in the scattering process which are influenced by the correlation between the localized spin and the electrode (equ. 2).” (lines 225-228).

We hope that this brief statement helps to follow the line of our model. Note, that both parts of our calculation, that is using rate equations as well as accounting for scattering processes up to third order has been very successfully used by many STM experiments. The new part is only the explicitly addressed influence of the spin density matrices by the correlation term.

1.6. Referee #1 writes: “...Note also that the definition of the total spin S is wrong”

While we are not defining the total spin S in our manuscript we believe that the referee is referring to the z -projection of the total magnetic moment m_z^{\uparrow} . For clarity, we change the text to: “...with different total magnetic moment in z -projection...” (lines 105-106)

The following additional changes have been made to the manuscript:

- All figures, mathematical equations, text and references have been formatted to comply with the Nature Communications style guide.
- The title was changed to: “Correlation driven transport asymmetries through coupled spins in a tunnel junction” to make it more specific.
- A data availability statement and a financial non-compete statement have been added.
- The figure captions of figure 2 and figure 4 we added a description for the error-bars: “The error-bars include the statistical and systematic error of the fit to the spectroscopic data.”.
- Additionally, several grammar, spelling, and punctuation errors were corrected.